# Caveolin-1 Alleviates Acetaminophen—Induced Hepatotoxicity in Alcoholic Fatty Liver Disease by Regulating the Ang II/EGFR/ERK Axis

**DOI:** 10.3390/ijms23147587

**Published:** 2022-07-08

**Authors:** Jiao Xin, Tingyu You, Xiangfu Jiang, Dongdong Fu, Jiarong Wang, Wei Jiang, Xiaowen Feng, Jiagen Wen, Yan Huang, Chengmu Hu

**Affiliations:** 1Anhui Province Key Laboratory of Major Autoimmune Diseases, Anhui Institute of Innovative Drugs, School of Pharmacy, Anhui Medical University, Hefei 230032, China; xj1782936538@163.com (J.X.); youtingyu906829@163.com (T.Y.); jiangxf123@163.com (X.J.); 17718291538@163.com (D.F.); 18895347881@163.com (J.W.); 13956960932@163.com (W.J.); fxiaow@126.com (X.F.); jiagen168@163.com (J.W.); aydhy@126.com (Y.H.); 2Institute for Liver Diseases of Anhui Medical University, School of Pharmacy, Anhui Medical University, Hefei 230032, China; 3Key Laboratory of Anti-Inflammatory and Immune Medicine, Ministry of Education, Hefei 230032, China

**Keywords:** alcoholic fatty liver disease, acetaminophen, hepatotoxicity, caveolin-1, angiotensin II, epidermal growth factor receptor

## Abstract

Acetaminophen (APAP) is a widely used antipyretic analgesic which can lead to acute liver failure after overdoses. Chronic alcoholic fatty liver disease (AFLD) appears to enhance the risk and severity of APAP-induced liver injury, and the level of angiotensin II (Ang II) increased sharply at the same time. However, the underlying mechanisms remain unclear. Caveolin-1 (CAV1) has been proven to have a protective effect on AFLD. This study aimed to examine whether CAV1 can protect the APAP-induced hepatotoxicity of AFLD by affecting Ang II or its related targets. In vivo, the AFLD model was established according to the chronic-plus-binge ethanol model. Liver injury and hepatic lipid accumulation level were determined. The levels of Angiotensin converting enzyme 2 (ACE2), Ang II, CAV1, and other relevant proteins were evaluated by western blotting. In vitro, L02 cells were treated with alcohol and oleic acid mixture and APAP. CAV1 and ACE2 expression was downregulated in APAP-treated AFLD mice compared to APAP-treated mice. The overexpression of CAV1 in mice and L02 cells alleviated APAP-induced hepatotoxicity in AFLD and downregulated Ang II, p-EGFR/EGFR and P-ERK/ERK expression. Immunofluorescence experiments revealed interactions between CAV1, Ang II, and EGFR. The application of losartan (an Ang II receptor antagonist) and PD98059 (an ERK1/2 inhibitor) alleviated APAP-induced hepatotoxicity in AFLD. In conclusion, our findings verified that CAV1 alleviates APAP-aggravated hepatotoxicity in AFLD by downregulating the Ang II /EGFR/ERK axis, which could be a novel therapeutic target for its prevention or treatment.

## 1. Introduction

Acetaminophen (APAP) is a widely used antipyretic analgesic, and in many countries it is an over-the-counter (OTC) drug. However, an overdose of APAP may lead to hepatocyte necrosis and acute liver failure [1,2]. As of today, N-acetylcysteine is the only approved drug to treat APAP-induced liver injury in patients after APAP overdose [3].

Alcoholic fatty liver disease (AFLD) is a major cause of morbidity and mortality worldwide, and it encompasses a spectrum of hepatic manifestations ranging from simple steatosis to steatohepatitis, cirrhosis, and hepatocellular carcinoma [4,5]. AFLD has become an important burden for public health as there is currently no drug for AFLD therapy.

Several studies have indicated that chronic AFLD enhances the risk and severity of APAP-induced liver injury [6]. As the early symptoms of AFLD are not obvious, it will greatly increase the probability of liver injury if you take APAP while don’t know that you have AFLD already. However, the molecular mechanism underlying APAP-induced pathogenesis of hepatic injury in AFLD remains to be elucidated. In order to explore this mechanism, in this study we used a chronic binge drinking model similar to the pathological manifestations of early AFLD in vivo, and used ethanol oleic acid mixture and L02 cells to establish a steatosis model in vitro, then used acetaminophen.

Caveolae are 50–100 nm vesicular membrane structures that are found in the plasma membranes of many cell types [7]. Caveolin-1 (CAV1) is the main component of caveolae, which is a multifunctional scaffolding protein with multiple binding partners that are associated with the regulation of lipid raft domains and numerous physiological and pathophysiological processes [8]. It functions in signal transduction [9], cholesterol homeostasis [9], and vesicular transport [10].

The recent outbreak of coronavirus disease 2019 (COVID-19) caused by the Severe Acute Respiratory Syndrome Coronavirus-2 (SARS-CoV-2) has resulted in a worldwide pandemic. Hepatic steatosis and Kupffer cell activation appear to be commonly encountered in the livers of SARS-CoV-2-infected deceased, together with vascular alterations. Thus there may be some connections between COVID-19 and hepatic steatosis. ACE2 is the main viral receptor for SARS-CoV and SARS-CoV-2 [11,12]. In addition, clathrin-dependent endocytosis is considered the primary endocytic route for SARS-CoV and caveolae-dependent endocytosis, an alternative to clathrin, has been also described as the endocytic route for some viruses, such as simian virus 40 (SV40) [13]. Therefore, we hypothesized if the mechanism of CAV1 alleviation of the APAP-induced hepatotoxicity in AFLD is relevant to ACE2 or its related pathways.

ACE2 is a member of the renin angiotensin system (RAS). There are two antagonistic axes in RAS: ACE-Ang II-AT1 axis and ACE2-Ang(1–7)-Mas axis. ACE 2 can convert Ang II into Ang (1–7) and antagonize the effect of Ang II. It has been reported that RAS can regulate glucose and lipid metabolism. The ACE/Ang II/AT1R axis can cause abnormal glucose metabolism in adipose tissue, while ACE2/Ang (1–7)/Mas axis can improve glucose metabolism in adipose tissue. It further shows that CAV1 can reduce the exacerbation of APAP induced hepatotoxicity in the AFLD state, which may be related to ACE2 or Ang II.

In addition, Ang II has been considered as a major contributor to the development and progress of NAFLD in recent years [14]. A prospective study showed that Ang II levels were significantly elevated in patients with NAFLD, and elevated Ang II levels were an independent risk factor for NAFLD [15]. In addition, a large number of studies also found that Ang II blockers such as angiotensin converting enzyme inhibitors (ACEI) and angiotensin receptor blockers (ARBs) can reduce the occurrence of NAFLD [16,17,18,19,20]. In conclusion, Ang II is closely related to the occurrence of steatosis. Blocking Ang II helps to reduce steatosis. However, its mechanism remains to be solved.

The purpose of this study was to explore whether CAV1 can improve the steatosis and liver injury aggravated by APAP in AFLD by regulating the Ang II related pathway. The study then explored the relationship between the phosphorylation of EGFR and ERK1/2 (the downstream of Ang II) and the aggravation of liver injury in alcoholic fatty liver induced by APAP, and this is the first study to link this disease model with Ang II/EGFR/ERK signaling pathways.

## 2. Results

### 2.1. APAP Aggravated Lipid Deposition and Liver Injury in Ethanol-Induced AFLD Mice

The body weight of mice was counted after randomized grouping. With the increase in modeling time, the body weight of mice in the APAP group saw no significant change, the EtOH group was lower than that of mice in the N group, and the body weight of mice in the EtOH group was further reduced after APAP treatment compared to that of mice in the APAP group (Figure 1B). The liver index reflects the degree of liver injury and fat accumulation in this organ. In the present study, the liver index in the APAP and EtOH groups was slightly higher than that in the N group; however, for mice in the EtOH + APAP group the liver index increased significantly (Figure 1C). The activities of ALT and AST and the content of TG and TC in the APAP and EtOH groups were higher than those in the N group. Compared to that found in the APAP group, the above parameters were significantly higher in the EtOH + APAP group (Figure 1D,E). The results of H&E staining were consistent with the above results (Figure 1F). The analysis of TG and TC, as well as H&E staining, indicated that lipid had accumulated. Fatty liver is usually accompanied by oxidative stress. Subsequently, we examined changes in oxidative stress levels. The activities of GSH and SOD and the content of MDA are classic indicators of oxidative stress. Biochemical measurements revealed that the activities of GSH and SOD were lower, while MDA content was higher in the APAP and EtOH group than in the N group. These parameters changed more significantly in the EtOH + APAP group than in the APAP group (Figure 1G–I). These results indicated that APAP aggravates lipid deposition and liver injury and induces an increased oxidative stress level in AFLD.

### 2.2. The Protein Expression of CAV1 and ACE2 Were Decreased While That of Ang II Was Increased in APAP-Treated AFLD Mice

CAV1 plays a key role in the pathogenesis of AFLD. In our previous studies, CAV1 expression was downregulated in both AFLD and non-alcoholic fatty liver disease (NAFLD) mice, and this downregulation probably contributed to an increased susceptibility to APAP in these mice [21,22]. Consistently, in the present study, the level of CAV1 was notably decreased in APAP-treated ethanol-feeding mice than APAP-treated alone (Figure 1J). In addition, we found that the ACE2 level was decreased while the Ang II level was increased (Figure 1J). The immunohistochemical results of Ang II were consistent with these findings (Figure 1K). Ang II is reported to result in oxidative stress, which is consistent with the above results. Therefore, ACE2 and Ang II may be key players in the pathogenesis of APAP and ethanol-induced liver injury.

### 2.3. CAV1 Reduced Lipid Deposition and Liver Injury Caused by APAP in AFLD and Decreased the Level of Oxidative Stress

The body weight and liver index showed no obvious change after LV-CAV1 was injected compared to the EtOH + APAP + LV-Control group (Figure 2A,B), but the content of TG and TC and the activities of ALT and AST in the EtOH + APAP + LV-CAV1 group were reduced (Figure 2C–F). Consistent with these findings, the EtOH + APAP + LV-Control group exhibited severe intrahepatic necrosis and extensive lipid droplets, but treatment with LV-CAV1 reduced the severity of these conditions (Figure 2G). We measured the protein levels of CAV1, and there was almost no difference between the N group and the LV-Control group, while the level of LV-CAV1 was significantly higher than that of the LV-Control group (Figure 2H). It shows that lentivirus transfection is successful. Moreover, we found that LV-CAV1 reversed ethanol- and APAP-induced elevated levels of oxidative stress (Figure 2I–K). These results indicated that LV-CAV1 effectively reduced APAP-induced hepatotoxicity and oxidative stress levels in EtOH.

### 2.4. CAV1 May Alleviate APAP-Induced Lipid Accumulation in AFLD by Affecting the Ang II/EGFR/ERK Axis and Restoring Autophagic Flux

To study this mechanism, we measured the protein levels of CAV1, ACE2, and Ang II. Immunohistochemical results showed that the number of Ang II-positive cells was increased in the EtOH + APAP + LV-Control group and decreased in the EtOH + APAP + LV-CAV1 group (Figure 3A). Western blotting results showed that the CAV1 and ACE2 levels were increased, and the Ang II level was decreased in the EtOH + APAP + LV-CAV1 group, in contrast to those in the EtOH + APAP+ LV-Control group (Figure 3B). Ang II combined with its receptor Angiotensin II Type 1 Receptor (AT1R) induces oxidative stress and activates EGFR. It has also been reported that CAV1 has a negative regulatory effect on EGFR. In addition, immunofluorescence results showed that there was an interaction between CAV1 and Ang II as well as CAV1 and EGFR (Figure 3C,D). However, how CAV1, Ang II, and EGFR are regulated during APAP and their contribution to AFLD remain unclear.

Ang II combined with its receptor AT1R can induce oxidative stress and activate EGFR and downstream MAPK signaling cascades, resulting in autophagy injury, thrombosis, and vasoconstriction. Consistently, in the EtOH + APAP + LV-Control group, EGFR and its downstream MAPK/ERK1/2 were highly phosphorylated (Figure 4A), while western blot results indicated that autophagy was blocked (Figure 4B), and TEM revealed a lower number of autophagosomes and increased numbers of lipid droplets (LDs) (Figure 4C). After LV-CAV1 was injected, all of these trends were inverted (Figure 4).

In conclusion, CAV1 may alleviate APAP-induced lipid accumulation in AFLD mice by affecting the Ang II/EGFR/ERK axis and restoring autophagic flux.

### 2.5. APAP Aggravated Lipid Deposition and Liver Injury Induced by Alcohol and Oleic Acid in L02 Cells

To study the specific mechanism, we constructed a mixed model of hepatic steatosis with alcohol and oleic acid on L02 cells in vitro. As shown in Figure 5A, orange lipid droplets significantly increased in the EtOH/OA + APAP group compared to the APAP group (Figure 5A). TG levels were also significantly upregulated after APAP treatment compared to those in the APAP group (Figure 5B). These results indicate that APAP combined with an EtOH/OA mixture aggravated lipid deposition in L02 cells. Consistent with the vivo experimental results, the protein expression of CAV1 and ACE2 was decreased while that of Ang II was increased after the EtOH/OA and APAP co-treatment compared to that of the APAP -treated L02 cells (Figure 5C).

### 2.6. CAV1 Alleviated Lipid Accumulation and Oxidative Stress via Suppressing Ang II/EGFR/ERK Signaling and Restoring Autophagic Flux in L02 Cells

We infected L02 cells with LV-CAV1 to overexpress CAV1. Western blotting indicated that the protein levels of CAV1 showed almost no difference between the N group and the LV-Control group, while the level of LV-CAV1 was significantly higher than that of the LV-Control group (Figure 6A). We found that upregulation of CAV1 reduced lipid droplet accumulation and ROS content (Figure 6B,D). Consistent with the vivo experiment, western blotting indicated that the level of ACE2 was decreased, while Ang II increased in the LV-Control group (Figure 6C). Interestingly, the changes were reversed when CAV1 was overexpressed (Figure 6B–D). Moreover, in the LV-Control group, EGFR and ERK were significantly activated and autophagy was inhibited (Figure 6E). After LV-CAV1 infection, phosphorylation of EGFR and ERK decreased significantly and autophagic flux was restored (Figure 6E). In summary, CAV1 may prevent AFLD development by suppressing Ang II/EGFR/ERK signaling and promoting autophagy to accelerate lipid metabolism.

### 2.7. CAV1 Silencing Exacerbated Lipid Accumulation and Activated the Ang II/EGFR/ERK Signaling in L02 Cells Induced by EtOH/OA and APAP

To further verify the protective effect of CAV1, we inhibited the expression of CAV1 by transfection with CAV1-siRNA. Western blotting results showed that CAV1 protein expression declined markedly in the CAV1-siRNA group compared to the Control-siRNA group (Figure 7A). The ORO staining results showed that the number of orange lipid droplets increased significantly after CAV1 was inhibited (Figure 7B). In addition, we found that when CAV1 was suppressed, ACE2 protein expression decreased, while Ang II protein expression increased (Figure 7C) and ROS levels increased significantly (Figure 7D). Western blotting results showed that the ratio of phosphorylated EGFR to EGFR and phosphorylated ERK to ERK increased significantly, while autophagy was inhibited when CAV1 expression was suppressed (Figure 7E). These results indicated that CAV1 deficiency may aggravate liver injury by activating Ang II/EGFR/ERK signaling and inhibiting autophagy.

### 2.8. Inhibition of Ang II and ERK1/2 Alleviated Lipid Accumulation and Restored Autophagic Flux in L02 Cells

To verify whether CAV1 prevents APAP-aggravated hepatic steatosis by suppressing Ang II/EGFR/ERK signaling and promoting autophagy, we applied two inhibitors to L02 cells.

One is losartan, the first oral nonpeptide Ang II receptor antagonist, which has been approved for use in 93 countries to treat hypertension. The existing literature has reported that losartan has a renal antioxidative effect and a protective effect on fatty liver [23,24]. However, the underlying mechanism is not clear. ORO staining revealed that losartan reduced the number of orange LDs (Figure 8A). Western blotting results showed that ACE2 expression was recovered and Ang II expression was suppressed, and the ratio of phosphorylated EGFR to EGFR also decreased (Figure 8B). These results were consistent with our expectations.

Another is PD98059, an ERK inhibitor, as we hypothesized that the effect of EGFR inhibition on hepatic steatosis may be mediated by MEK/ERK. Using PD98059, we found that the liver injury improved (Figure 8C). Consistent with this finding, p-ERK1/2 expression was greatly reduced with PD98059 in the EtOH/OA + APAP group compared to that in the control groups (Figure 8D). Altogether, these data revealed that EGFR signaling regulates hepatic steatosis via the MEK/ERK cascade.

## 3. Discussion

Alcoholic fatty liver disease is a chronic disease with high incidence [25]. It is typically characterized by hepatocyte injury and excessive lipid accumulation in the hepatocytes. Many host and environmental factors and comorbidities have been shown to modify the development and progression of AFLD, including age, sex, genetic factors, drinking pattern, obesity, and chronic viral hepatitis. Among these risk factors, ethanol intake is a known risk factor for various types of chronic liver diseases, including AFLD.

Acetaminophen is a widely used OTC drug for the treatment of fever and pain. It has been reported that obesity-related hepatic lesions appear to increase the risk and severity of APAP-induced liver damage [6]. Growing evidence has also shown that oxidative stress contributes to hepatic histopathological changes by promoting the formation of lipid peroxides and decreasing the levels of SOD and GSH [26]. In addition, studies have shown that many toxic substances-induced liver injuries can be significantly reduced by targeting oxidative stress and inflammation. In acrylamide-mediated rat liver injury, thymequinone (TQ), the active component of Nigella sativa, exerted a liver protective effect through its antioxidant properties [27]. Limonin also ameliorated APAP-induced hepatotoxicity by activating the Nrf2 antioxidant pathway and inhibiting the NF-κB inflammatory response [28].

Caveolae are a subdomain of lipid rafts that are defined as “small (10–200 nm) heterogeneous membrane domains enriched in sterol and sphingolipids that are involved in the compartmentalization of various cellular processes” [29]. CAV1 is a 21- to 24-kDa molecule that is an essential structural component of caveolae membranes in vivo and participates in vesicular trafficking events and transduction processes. It plays a regulatory role in several signaling pathways, such as those involving Src family tyrosine kinase, G-proteins, epidermal growth factor receptor, protein kinase C, and endothelial nitric oxide synthase. In addition, our previous studies have shown that alcohol-induced fatty liver mice are more susceptible to APAP-induced liver injury, and that CAV1 seems to have a protective effect on APAP and alcohol consumption [21,22]. However, how such regulation during CAV1 affects APAP and AFLD pathogenesis remains elusive. Therefore, in the present study, we used a chronic-plus-binge ethanol model that is similar to the pathological manifestation of early stage AFLD to explore this mechanism, while an alcohol and oleic acid mixture and L02 cells were used to establish a steatosis model in vitro.

We found that the ACE2 level was decreased while the Ang II level was increased in APAP-induced liver injury in AFLD. ACE2 and Ang II may be key players in AFLD pathogenesis. Ang II has been reported to induce oxidative stress and therefore we examined changes in oxidative stress levels. Biochemical measurements indicated that APAP induced an increase in the level of oxidative stress in AFLD indeed. In addition, immunofluorescence results evidenced an interaction between CAV1 and Ang II. However, how ACE2, Ang II, and CAV1 are regulated during APAP and their contribution to AFLD remain unclear.

Angiotensin II combined with its receptor AT1R activates EGFR, a prototypical tyrosine kinase receptor of the ErbB family, which is highly expressed in the liver and known to be involved in hepatocyte proliferation, liver regeneration, and hepatocellular carcinogenesis [30,31]. EGFR can be activated by a variety of extracellular ligands, including EGF, transforming growth factor alpha (TGF-α), heparin-binding EGF (HB-EGF), and amphiregulin [32]. Ligand binding leads to the dimerization of EGFR and the auto-phosphorylation of tyrosine residues in its cytoplasmic domain, which then acts as a hub for numerous cell signaling pathways. The binding of EGF1 to its receptor EGFR initiates a kinase cascade that results in the activation of MAPK [33,34]. It was recently reported that EGF receptors are highly enriched in the caveolae-rich fractions of human fibroblasts [35]. EGF bound to receptors in caveolae stimulates the recruitment of Raf-1 to this domain, activating the Ras-Raf-1-MAP2K/MEK-MAPK/ERK pathway. In addition, CAV1 has been shown to negatively regulate EGFR [36]. Several studies have shown that the activation of ERK can inhibit autophagy [37,38,39]. In our previous studies, CAV1 was shown to inhibit autophagy [22]. Thus, in the present study, we investigated whether CAV1 could alleviate APAP-aggravated lipid accumulation in AFLD by regulating the Ang II/EGFR/ERK axis, ameliorating oxidative stress and restoring autophagic flux.

Consistently, in the EtOH + APAP + LV-Control group, EGFR and its downstream MAPK/ERK were highly phosphorylated. The TEM results revealed a lower number of autophagosomes in this group, and western blotting results indicated that autophagy was blocked. After LV-CAV1 was injected, all these results were inverted. EGFR inhibitors have been reported to have protective effects against liver injury and lipid deposition [34,35]. EGFR signaling functions via three downstream mediators, MEK/ERK, PI3K/AKT, and JAK/STAT3. We found that the ratio of phosphorylated ERK to ERK increased, while autophagy was inhibited when CAV1 expression was suppressed. These results indicated that CAV1 deficiency may aggravate liver injury by activating EGFR/ERK signaling and inhibiting autophagy.

Subsequently, we applied two inhibitors to the L02 cells. Losartan, the first oral non-peptide Ang II receptor antagonist, has been reported to have a protective effect on fatty liver. Our results showed that losartan reduced the number of orange fat drops, recovered ACE2 expression, suppressed the expression of Ang II, and phosphorylated EGFR. Phosphorylation of ERK has been reported to inhibit autophagy, and our previous studies have demonstrated that promoting autophagy improves APAP-aggravated lipid deposition and liver injury [22,40]. We used the ERK1/2 inhibitor PD98059, to inhibit the activation of phosphorylated ERK1/2. It was observed that lipid deposition was alleviated and the ratio of phosphorylated ERK to ERK declined with PD98059, while autophagic flux was restored. This suggested that the inhibition of ERK1/2 phosphorylation results in the recovery of autophagy.

In conclusion, the current study revealed that CAV1 alleviates APAP-induced lipid accumulation in AFLD by regulating the Ang II/EGFR/ERK axis, ameliorating oxidative stress, and restoring autophagic flux (Figure 9). Given the beneficial effects of CAV1, Ang II inhibitor, and ERK1/2 inhibitor on diseased livers, we suggest that these might be potential therapeutic strategies for patients with APAP-induced lipid accumulation in AFLD. In the future, drugs targeting CAVI may be designed to improve the expression activity of CAV1 and prevent the occurrence and development of related diseases. Moreover, a clinical study confirmed that serum CAV1 levels in heavy drinkers was negatively correlated with the degree of alcoholic liver injury [41], which makes the study of CAV1 more meaningful, especially for patients with alcoholic fatty liver who need acetaminophen to relieve fever and pain.

The study is the first to link APAP-induced lipid accumulation in AFLD with Ang II/EGFR/ERK signaling pathways. This may provide a new possible therapeutic strategy for its treatment. Despite the outcomes, our experiments had some limitations. In addition to detecting the protein expression of ACE2 and Ang II, the expression of other related proteins such as Ang (1–7), MasR, and AT1R should also be detected, and changes in the levels of these genes should be determined. Moreover, the other two downstream mediators of EGFR (PI3K/AKT and JAK/STAT3) should be studied to determine the underlying mechanism.

## 4. Materials and Methods

Interventionary studies involving animals or humans, and other studies that require ethical approval, must list the authority that provided approval and the corresponding ethical approval code.

### 4.1. Animal Experiments

Seven-week-old C57BL/6J male mice (18–20 g) were obtained from the Laboratory Animal Center of Anhui Medical University, Hefei, China. All animals were raised under standard conditions (24 ± 2 °C, 12-hour day/night cycle). The study was conducted in accordance with the requirements of the Animal Care and Ethics Committee of Anhui Medical University (NO: LLSC20190279). The AFLD model was established according to the chronic-plus-binge ethanol model [42]. The total feeding time was 21 days. After five days of adaptive feeding, a CAV1 overexpression mouse model was constructed by tail vein injection of lentivirus (LV) overexpressing CAV1, which was designed and chemically synthesized by Shanghai GeneChem Co. Ltd. (Shanghai, China). Adaptive feeding continued for one week. From the sixth day onwards, all animals were randomly divided into 12 groups (*n* = 8 per group): (1) normal group (N), (2) APAP group (APAP), (3) ethanol-feeding group (EtOH), (4) EtOH with APAP group (EtOH + APAP), (5) N with LV control (LV-Control) group (N + LV-Control), (6) APAP with LV-Control (APAP + LV-Control), (7) EtOH with LV-Control (EtOH + LV-Control), (8) EtOH with APAP and LV-Control (EtOH + APAP + LV-Control), (9) N with LV-CAV1 (N + LV-CAV1), (10) APAP with LV-CAV1 (APAP + LV-CAV1), (11) EtOH with LV-CAV1 (EtOH + LV-CAV1), and (12) EtOH with APAP and LV-CAV1 (EtOH + APAP + LV-CAV1). The EtOH groups were fed an alcohol liquid diet (TP4030A, TROPHIC, Nantong, China), whereas the other groups were fed a normal diet (TP4030C; TROPHIC, Nantong, China). On the last day of the experiment, mice were administered APAP (280 mg/kg) or an equal volume of phosphate buffered saline (PBS) by gavage (only once). The APAP solution was dissolved in PBS. Nine hours before sacrifice, alcohol (5 g/kg) was administered via the intragastric route. After fasting for 24 h, blood and liver tissue samples were collected for blood biochemical and histopathological analyses (Figure 1A).

### 4.2. Biochemical Measurements

Serum alanine aminotransferase (ALT), aspartate aminotransferase (AST), triglycerides (TG), and triglycerides (TC) were measured using assay kits (Jiancheng, Nanjing, China) and a microplate reader (Biotek, Winooski, VE, USA).

Portions of liver tissue were homogenized and dissolved in PBS to measure the glutathione (GSH) activity, lipid peroxidation, malondialdehyde (MDA) content, and superoxide dismutase (SOD) activity using the Total SOD Assay Kit with WST-8 (Beyotime, Shanghai, China) according to the manufacturer’s instructions.

### 4.3. Histopathologic Examination of Liver Tissue

Mouse liver samples were stored in 4% paraformaldehyde, fixed for 48 h, embedded in paraffin, cut into thin slices (5 µm), and stained with hematoxylin and eosin (H&E).

### 4.4. Transmission Electron Microscopy (TEM)

Mouse liver sections were fixed for 2 h at 4 °C with 4% glutaraldehyde solution and for 1 h at 4 °C with 1% osmium tetroxide solution. The sections were embedded in epoxy resin and cut using an ultramicrotome. The ultrathin sections (50–70 nm) were stained, and the images were observed by TEM (Hitachi, Tokyo, Japan).

### 4.5. Cell Culture and Treatment

Human normal hepatocyte L02 cells were obtained from the Cell Bank of the Chinese Academy of Sciences (Shanghai, China). Cells were grown and maintained in Dulbecco’s modified Eagle’s medium (DMEM; Gibco, Thermo Fisher Scientific, Waltham, MA, USA) supplemented with 10% FBS (EVERY GREEN, Hangzhou, China) and 1% penicillin/streptomycin at a constant temperature of 37 °C in a 5% CO2 incubator. An in vitro model of hepatocyte steatosis was established by co-treating L02 cells with 87 mM ethanol and 100 µM oleic acid (EtOH/OA) for 48 h. APAP (4 mM) was added during the last 24 h, and the plate was incubated for another 24 h. The cells were divided into four groups: (1) normal group (N), (2) APAP group (APAP), (3) ethanol and oleic acid group (EtOH/OA), and (4) EtOH/OA with APAP (EtOH/OA + APAP).

### 4.6. Cell Transfection and Inhibitor Administration

Lentivirus overexpressing CAV1 (LV-CAV1) was designed and chemically synthesized by Shanghai GeneChem Co. Ltd. (Shanghai, China). To overexpress CAV1 in vitro, L02 cells were transfected with LV-CAV1 or LV-Control using 5 μg/mL polybrene (Shanghai GeneChem Co. Ltd.) according to the manufacturer’s protocol. After transfection for 6 h, the medium was replaced with fresh medium, and the cells were further treated with EtOH/OA and APAP. To determine whether Ang II was involved in CAV1-mediated liver protection, 1 µM losartan (an Ang II receptor antagonist) (Merck & Co., Inc., Kenilworth, NJ, USA) was administered 1 h before the EtOH/OA treatment. To determine whether extracellular signal-regulated protein kinase (ERK1/2) was involved in CAV1-mediated liver protection, 20 µM PD98059 (an ERK1/2 antagonist) (Merck & Co., Inc.) was administered 1 h before the EtOH/OA treatment. L02 cells were collected for western blotting and ORO staining.

Small interfering RNA (siRNA) for CAV1 and negative control siRNA were synthesized by Shanghai GenePharma Co., Ltd. (Shanghai, China). The sequences were as follows: CAV1-siRNA, 5-CCGCAUCAACUUGCAGAAATT-3 and 5-UUUCUGCAAGUUGAUGCGGTT-3; and control siRNA, 5-UUCUCCGAACGUGUCACGUTT-3 and 5-ACGUGACACGUUCGGAGAATT-3′. L02 cells were transfected with CAV1-siRNA and Control-siRNA for 6 h using Lipofectamine 3000 reagent (Polyplus-jetPRIME, Illkirch, France) according to the manufacturer’s protocol. After transfection for 6 h, the medium was replaced with fresh medium, and the cells were further treated with EtOH/OA and APAP. The cells were then harvested for the follow-up experiments.

### 4.7. ORO Staining and Measurement of Intracellular TG

The cells were seeded into 6-well plates and cultured with designated reagents and drugs, fixed with 4% paraformaldehyde, and stained with a freshly prepared working solution of ORO at room temperature. Thirty minutes later, the oil-red dye is removed and excess dye is washed away until the background is nearly colorless. After 2 h, orange-red lipid droplets were observed in the cells under the bright field of a fluorescence inverted microscope (Olympus, Tokyo, Japan). The content of TG and TC in cells was measured using assay kits (Jiancheng, Nanjing, China) and calculated according to the manufacturer’s instructions.

### 4.8. Detection of Reactive Oxygen Species (ROS) Production

The cell-permeable fluorophore dihydroethidium (DHE) was used to evaluate ROS production. DHE is oxidized by ROS to a novel product that binds to DNA, thereby enhancing intracellular fluorescence. Images were captured using a fluorescence microscope at 400× magnification (excitation and emission wavelengths of 490 and 610 nm, respectively). The fluorescence intensity values were quantified using ImageJ software (National Institutes of Health, Bethesda, MD, USA).

### 4.9. Western Blot Analysis

Total proteins from liver tissues and L02 cells were extracted using a RIPA lysis buffer containing 1% protease and phosphatase inhibitors (Solarbio, Beijing, China). The lysates were separated using sodium dodecyl sulfate-polyacrylamide gel electrophoresis (SDS-PAGE) after determining protein concentration using a BCA protein assay kit (Beyotime). Subsequently, the proteins were transferred onto polyvinylidene difluoride (PVDF) membranes (Millipore Corp, Billerica, MA, USA), which were then blocked with 5% non-fat milk and incubated with primary antibodies at 4 °C overnight. After the membranes were washed with Tris-buffered saline plus 0.1% Tween^®^ 20 Detergent (TBST), they were incubated with peroxidase-labeled secondary antibody (1:5000, ZS-GB, Beijing, China) at 37 °C for 1 h. Protein blots were detected using an enhanced chemiluminescence kit (ECL-plus, Thermo Fisher Scientific) and quantified using ImageJ software. The primary antibodies used in this study were ACE2, CAV1 (Proteintech, Wuhan, China), P62 (Abcam, Cambridge, UK), Ang II, β-actin (Bioss, Beijing, China), LC3 (Santa Cruz Biotechnology, Inc., Dallas, TX, USA), EGFR (Cell Signaling Technology, Inc., Danvers, MA, USA), p-EGFR, ERK1/2, and p-ERK1/2 (Wanleibio, Shenyang, China).

### 4.10. Statistical Analyses

The data were analyzed by Student’s t-test or one-way analysis of variance (ANOVA) followed by Tukey’s post hoc tests using GraphPad Prism 6.0 and SPSS 18.0 (IBM, Armonk, NY, USA). For all other data, a one-way ANOVA with Bonferroni’s multiple comparisons test was performed. Data represent the mean ± SEM for eight mice or three independent experiments. The statistical significance was set at *p* < 0.05.

## 5. Conclusions

The current study revealed that CAV1 alleviates APAP-induced lipid accumulation in AFLD by regulating the Ang II/EGFR/ERK axis, ameliorating oxidative stress, and restoring autophagic flux (Figure 9).

## Figures and Tables

**Figure 1 ijms-23-07587-f001:**
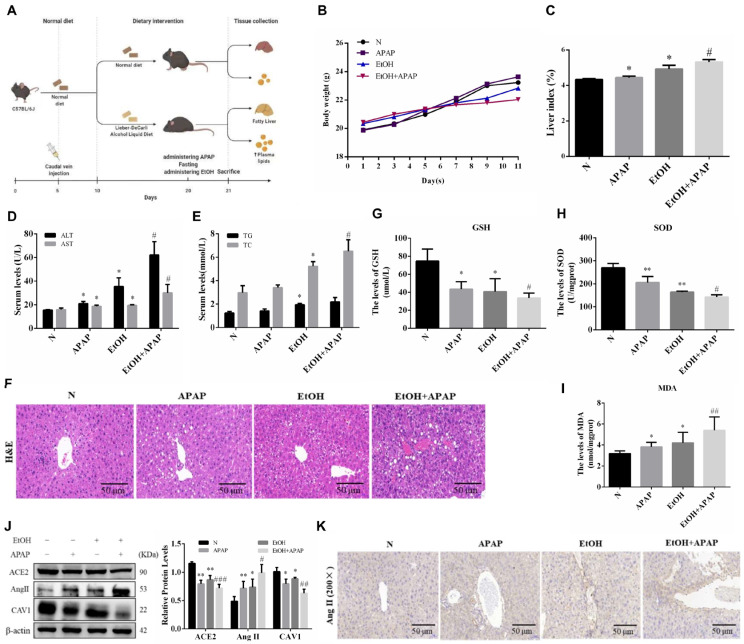
APAP aggravated liver injury in ethanol-induced AFLD mice; the protein expression of CAV1 and ACE2 were decreased while the Ang II was increased in APAP treated AFLD mice, with the oxidative stress levels increasing. Male C57BL/6J mice were fed an alcohol liquid diet or equivalent liquid control diet, while APAP (280 mg/kg) was given by gavage on the last day. (**A**) Animal experiment process. (**B**) The bodyweight of mice in each group. (**C**) The liver index of mice in each group. (**D**) The serum activity of ALT and AST. (**E**) The serum activity of TG and TC. (**F**) Representative images of hematoxylin and eosin staining results (Scale bar = 50 μm). (**G**) Serum glutathione (GSH). (**H**) Liver superoxide dismutase (SOD) activity. (**I**) Malondialdehyde (MDA) levels. (**J**) Western blot analysis of the relative protein levels of CAV1, ACE2 and Ang II were measured in liver tissue. (**K**) Representative images of immunohistochemistry results of Ang II for liver tissue sections (Scale bar = 50 μm). Data represent the mean ± SEM (*n* = 8 mice). * *p* < 0.05 vs. N group; ** *p* < 0.01 vs. N group; # *p* < 0.05 and ## *p* < 0.01 vs. APAP group. ### *p* < 0.001.

**Figure 2 ijms-23-07587-f002:**
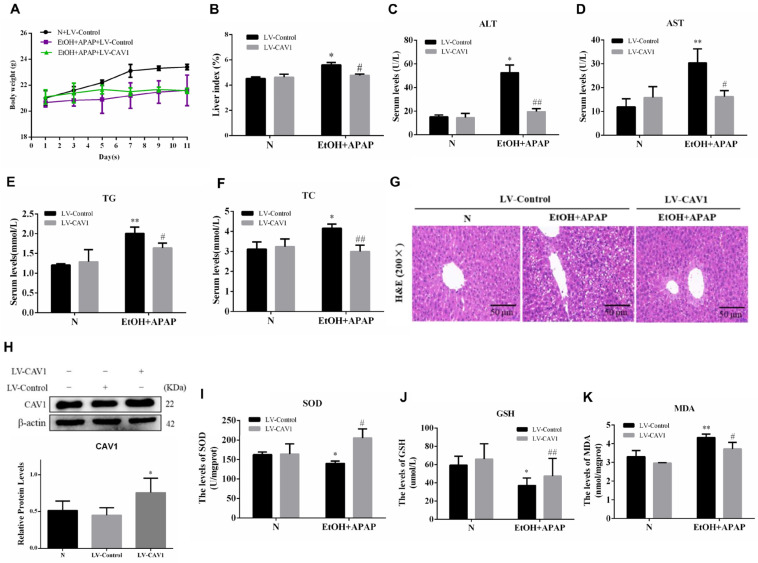
CAV1 reduced lipid deposition and liver injury caused by APAP in AFLD and decreased the level of oxidative stress. (**A**) The bodyweight of mice in each group. (**B**) The liver index of mice in each group. (**C**) The serum activity of ALT. (**D**) The serum activity of AST. (**E**) The serum activity of TG. (**F**) The serum activity of TC. (**G**) Representative images of hematoxylin and eosin staining results (Scale bar = 50 μm). (**H**) Western blot analysis of the relative protein levels of CAV1. (**I**) Serum glutathione (GSH). (**J**) Liver superoxide dismutase (SOD) activity. (**K**) Malondialdehyde (MDA) levels. Data represent the mean ± SEM (*n* = 8 mice). * *p* < 0.05 vs. N + LV-Control group; ** *p* < 0.01 vs. N + LV-Control group; # *p* < 0.05 and ## *p* < 0.01 vs. EtOH + APAP + LV-Control group.

**Figure 3 ijms-23-07587-f003:**
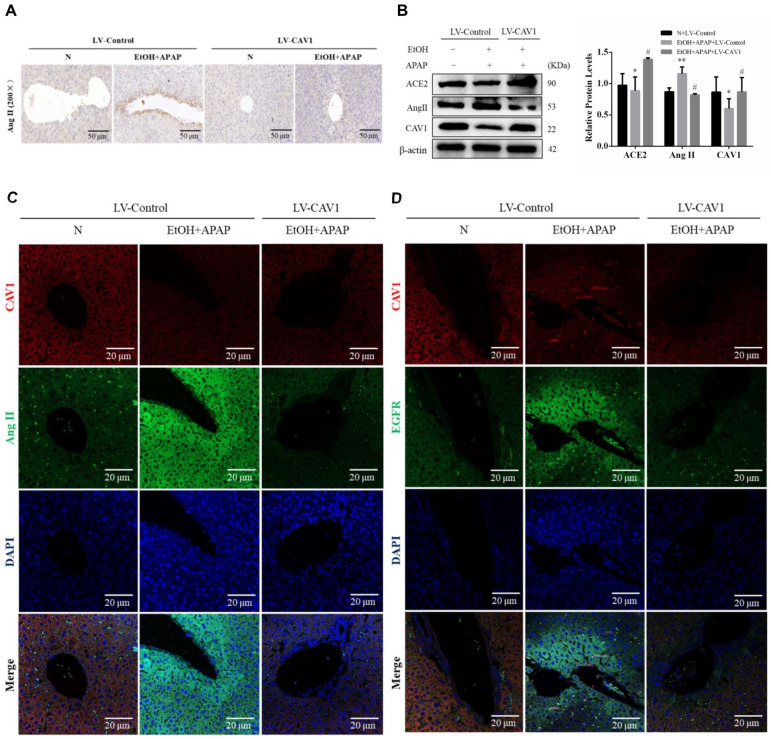
The protein expression of Ang II was inhibited after CAV1 overexpression. (**A**) Representative images of immunohistochemistry results of Ang II for liver tissue sections (Scale bar = 50 μm). (**B**) Western blot analysis of the relative protein levels of CAV1, ACE2 and Ang II were measured in liver tissue. (**C,D**) Representative images of immunofluorescence double staining results of CAV1 and Ang II as well as CAV1 and EGFR. DAPI was used for nuclear staining (blue). Red color represents CAV1 staining. Green color represents Ang II or EGFR staining (Scale bar = 20 μm). Data represent the mean ± SEM (*n* = 8 mice). * *p* < 0.05 vs. N + LV-Control group; ** *p* < 0.01 vs. N + LV-Control group; # *p* < 0.05 vs. EtOH + APAP + LV-Control group.

**Figure 4 ijms-23-07587-f004:**
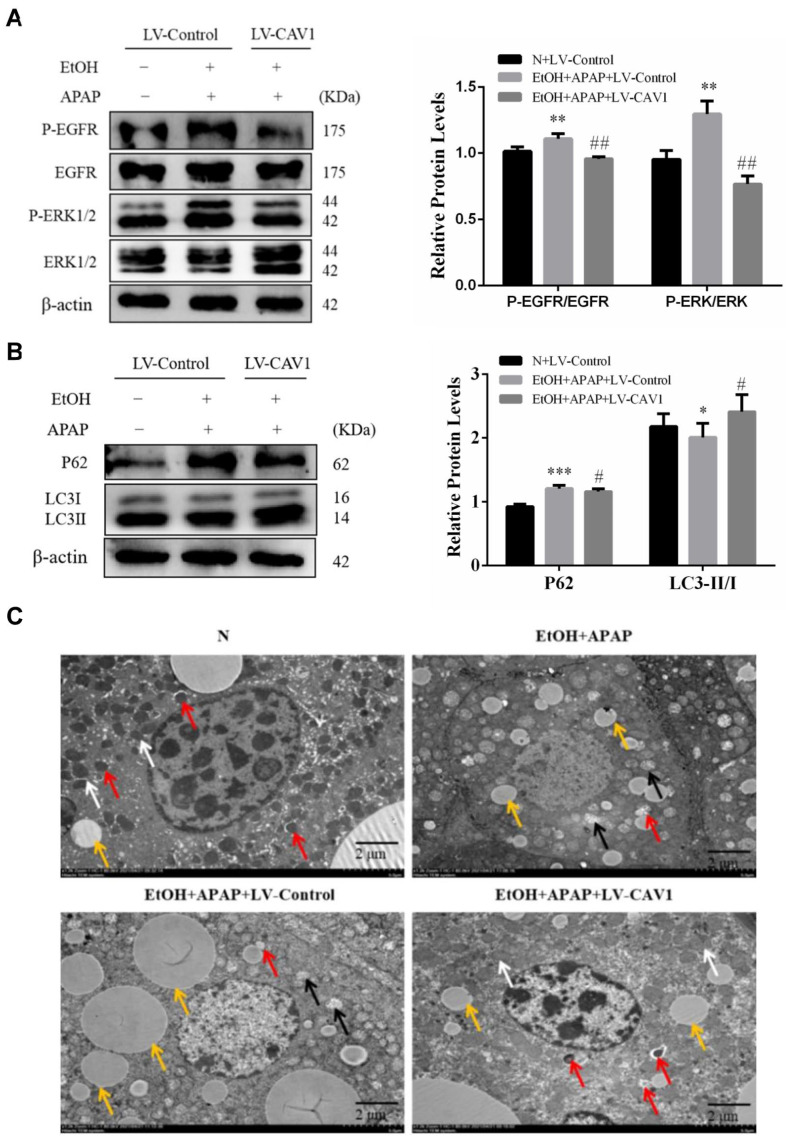
CAV1 alleviates APAP-induced lipid accumulation in AFLD by affecting the Ang II/EGFR/ERK axis and restoring autophagic flux. (**A**) Western blot analysis of p-EGFR/EGFR and p-ERK/ERK protein levels. (**B**) Western blot analysis of P62 and LC3 II / I protein levels. (**C**) Transmission electron microscopy analysis of mitochondria, autophagosomes and lipid droplets in liver tissues. White arrows show normal mitochondria, yellow arrows show damaged mitochondria, black arrows show lipid droplets and red arrows show autophagosomes. The number of autophagosomes per given area (mean ± S.D.) was determined (Scale bar = 5.0 μm). Data represent the mean ± SEM (n = 8 mice). * *p* < 0.05 vs. N + LV-Control group; ** *p* < 0.01 vs. N + LV-Control group; *** *p* < 0.001 vs. N + LV-Control group; # *p* < 0.05 and ## *p* < 0.01 vs. EtOH + APAP + LV-Control group.

**Figure 5 ijms-23-07587-f005:**
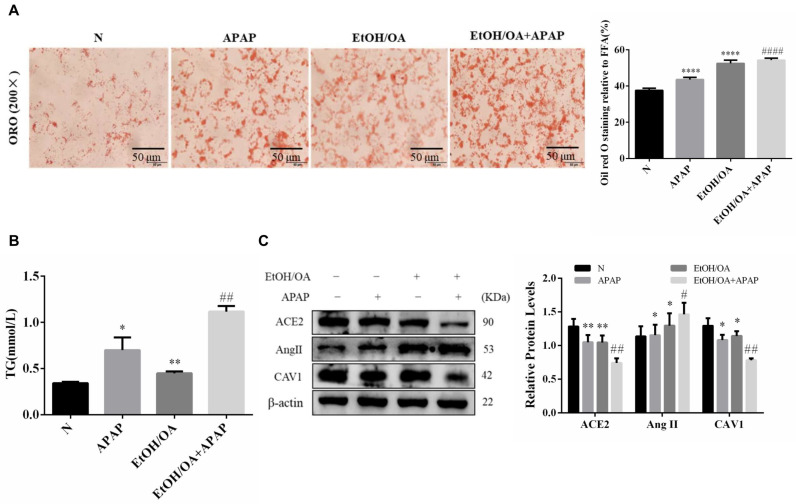
APAP aggravated lipid deposition and liver injury induced by alcohol and oleic acid in L02 cells, and the protein expression of the CAV1 and ACE2 were decreased while the Ang II was increased. (**A**) ORO staining in L02 cells (Scale bar = 50 μm). (**B**) The TG levels treated with EtOH/OA and APAP. (**C**) Western blot analysis of ACE2, Ang II and CAV1 protein levels. (n = 3 independent experiments). * *p* < 0.05 vs. N group; ** *p* < 0.01 vs. N group; **** *p* < 0.0001 vs. N group; # *p* < 0.05 vs. APAP group; ## *p* < 0.01 vs. APAP group; #### *p* < 0.0001 vs. APAP group.

**Figure 6 ijms-23-07587-f006:**
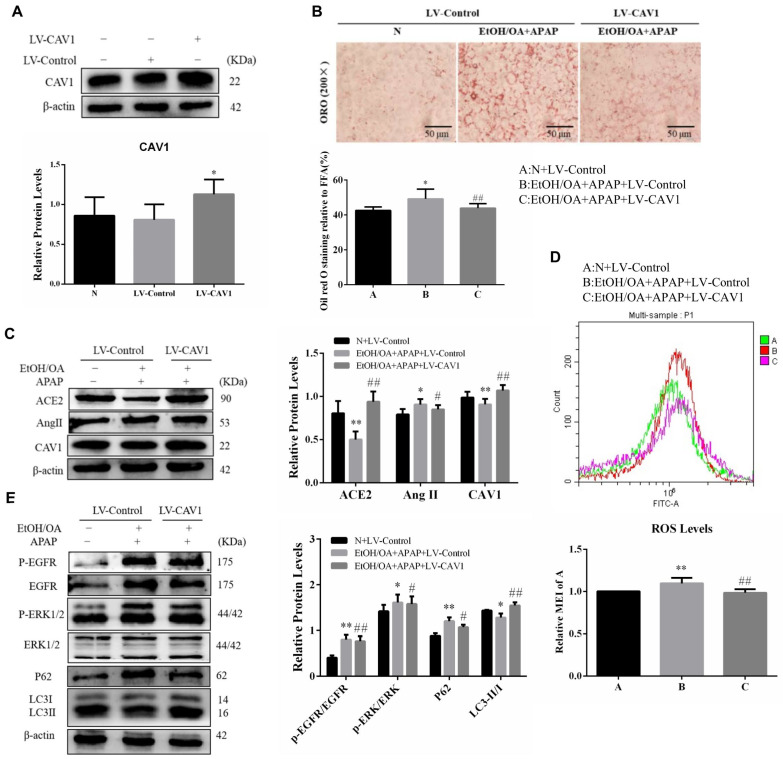
CAV1 alleviated lipid deposition, suppressed the Ang II/EGFR/ERK signaling and restored autophagic flux in EtOH/OA and APAP treated L02 cells. (**A**) Western blot analysis of CAV1 protein levels. (**B**) Representative images of ORO staining of lipid droplets in L02 cells (Scale bar = 50 μm). (**C**) Western blot analysis of ACE2, Ang II and CAV1 protein levels. (**D**) ROS levels. (**E**) Western blot analysis of p-EGFR/EGFR, p- ERK/ERK, P62 and LC3 II / I protein levels. Data represent the mean ± SEM (*n* = 3 independent experiments). * *p* < 0.05 vs. N + LV-Control group; ** *p* < 0.01 vs. N + LV-Control group; # *p* < 0.05 and ## *p* < 0.01 vs. EtOH/OA + APAP + LV-Control group.

**Figure 7 ijms-23-07587-f007:**
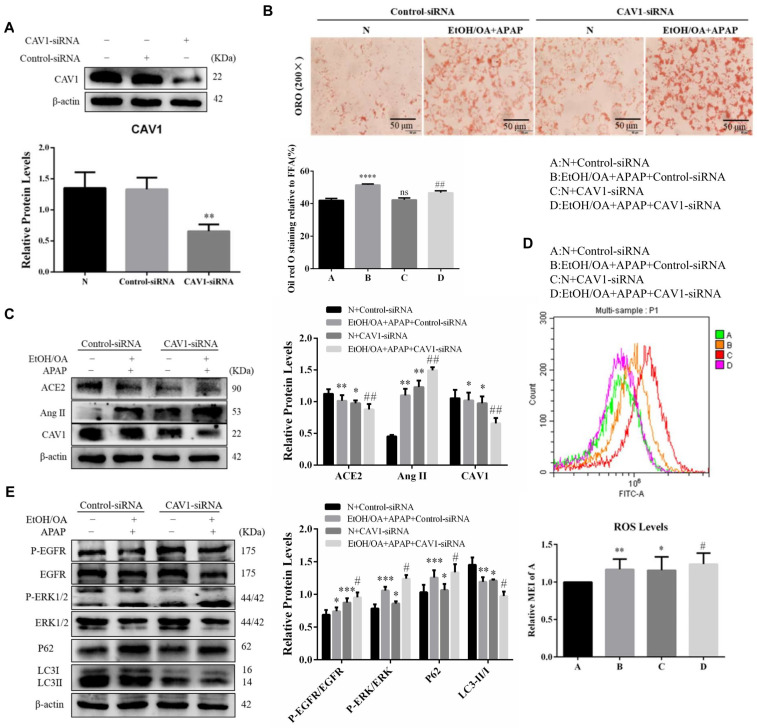
CAV1 silencing exacerbated lipid accumulation and activated the Ang II/EGFR/ERK signaling in L02 cells induced by EtOH/OA and APAP. (**A**) Western blot analysis of CAV1 protein levels. (**B**) Representative images of ORO staining of lipid droplets in L02 cells (Scale bar = 50 μm). (**C**) Western blot analysis of ACE2, Ang II and CAV1 protein levels. (**D**) ROS levels. (**E**) Western blot analysis of p-EGFR/EGFR, p- ERK/ERK, P62 and LC3 II / I protein levels. Data represent the mean ± SEM (n = 3 independent experiments). * *p* < 0.05 vs. N + Control-siRNA group; ** *p* < 0.01 vs. N + Control-siRNA group; *** *p* < 0.001 vs. N + Control-siRNA group; **** *p* < 0.0001 vs. N + Control-siRNA group; ^ns^
*p* > 0.05 vs. N + Control-siRNA group; # *p* < 0.05 and ## *p* < 0.01 vs. EtOH/OA + APAP + Control-siRNA group.

**Figure 8 ijms-23-07587-f008:**
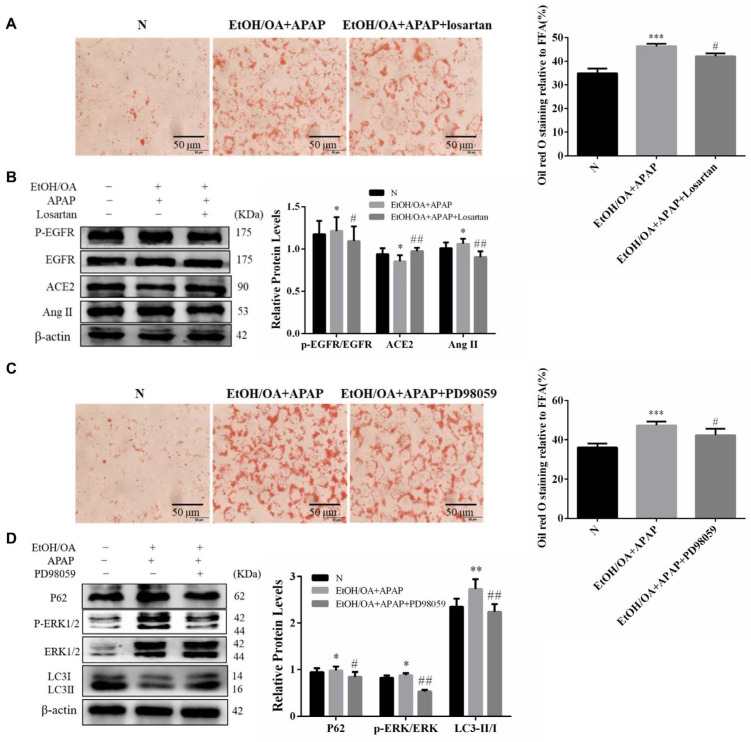
Inhibition of Ang II and ERK1/2 alleviated lipid accumulation and restored autophagic flux in L02 cells. Pretreatment with losartan or PD98059 one hour before cell modeling. (**A**) Representative images of ORO staining with losartan (2 μM) in L02 cells (Scale bar = 50 μm). (**B**) Western blot analysis of ACE2, Ang II and p-EGFR/EGFR protein levels. (**C**) Representative images of ORO staining with PD98059 (40 μM) in L02 cells (Scale bar = 50 μm). (**D**) Western blot analysis of p-ERK/ERK, P62 and LC3 II/I protein levels. Data represent the mean ± SEM (*n* = 3 independent experiments). * *p* < 0.05 vs. N group; ** *p* < 0.01 vs. N group; *** *p* < 0.001 vs. N group; # *p* < 0.05 and ## *p* < 0.01 vs. EtOH/OA + APAP group.

**Figure 9 ijms-23-07587-f009:**
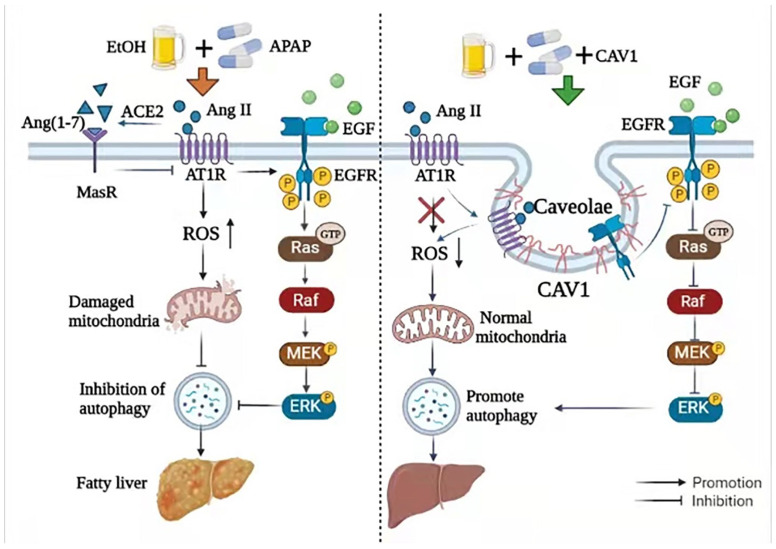
CAV1 alleviates APAP-induced lipid accumulation in AFLD by regulating the Ang II/EGFR/ERK axis, ameliorating oxidative stress, and restoring autophagic flux. **EtOH**, ethanol; **APAP**, acetaminophen; ACE2, angiotensin-converting enzyme 2; Ang II, angiotensin II; EGFR, epidermal growth factor receptor; MAPK/ERK, mitogen-activated protein kinase.

## Data Availability

The data presented in this study are available on request from the corresponding author.

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
