# Peer review of "Caveolin-1 Alleviates Acetaminophen—Induced Hepatotoxicity in Alcoholic Fatty Liver Disease by Regulating the Ang II/EGFR/ERK Axis"

_ijms, 2022, doi:10.3390/ijms23147587_

Round 1
Reviewer 1 Report
The authors investigated the effect of Caveolin on AAP+EtoH induced liver injury model in animals and cells. Although some of the results are impressive, they should improve MS according to comments. My detailed comments are as belows:
1. AAP and Ethanol was well known to synergistically induce liver injuty through metabolism mediated by CYP2E1. However, many cell lines don't express metabolizing enzymes. The authors should check CYP, ADH expression and activity in L02 cells
2. Oil red o staining was extracted with isopropanol and can be measured. The authors should measure oil red o staining intensity.
3.Fig 4 a and b, actin blot looks to be duplicated.
4. Some of western blot data were poor like Fig4 LC3 data. The authors should improve western blot data.
Reviewer 2 Report
Dear authors
Please see my comment bellow:
L85-87. Please make the aim of the study a separate, last paragraph of Introduction section (to make it easier visible for those interested in the topic), highlighting better some aspects by responding to the following questions: Which is the novelty of your study or the special aspects it brings to the field? What makes different your study from others in the same/similar topic, already published?
Results section. From Figure 1 to figure 8, all of them have inserted in the upper left corner Fig.no. Please crop the figures without it.
Also, please extend figures 4, 5, 6 and 9 on the entire width of the page. They are very small, in the actual form.
Please better develop the Discussion chapter by better detailing the clinical impact of your study. How can caveloin-1 activity be increased? How does ACEI medication influence acetaminophen toxicity? What other medication is available/or are experimentally tested that can reduce the negative impact of toxic substances on the liver activity? I suggest checking and referring to https://link.springer.com/article/10.1007/s11356-020-09516-3.
Also, a graphic/figure depicting caveolin activity would highlight this part.
L375. Figure 9. After its title, please explain each abbreviation used in the figure, according to the Instructions for author: “Acronyms/Abbreviations/Initialisms should be defined the first time they appear in each of three sections: the abstract; the main text; under the first figure or table; when defined for the first time, the acronym/abbreviation/initialism should be added in parentheses after the written-out form.”
After L375. Please add a paragraph describing the strengths and limitations of your study.
L391. I'm sure the authors know that in order to have a statistically significant statistic, each group must have at least 33 individuals/samples (whether it's experimental or clinical trials) - and in L391, the authors stated that” n = 8 per group“. For <33, statistic is not relevant. I am confused about the statistic that was performed. Please explain.
Conclusion section is missing. Please insert it, briefly highlighting (without numerical values), the main findings of your research.
Reviewer 3 Report
General comment-This is a clearly presented and well-written paper from Dr. Chengmu Hu’s group. In this manuscript, the authors examined the role of caveolin-1 in alleviating APAP-induced hepatotoxicity in alcoholic fatty liver disease (AFLD) by regulating the angiotensin II/EGFR/ERK pathways. AFLD affects the people worldwide. Its manifestations ranges from steatosis, NASH, cirrhosis to HCC. Currently, there is no drug for AFLD therapy. Overuse of over the counter drug acetaminophen (APAP, common analgesic) may lead to hepatocyte necrosis and liver failure. The following study demonstrate the mechanistic role of caveolin-1 in alleviating APAP-induced hepatotoxicity in AFLD by regulating the angiotensin II/EGFR/ERK axis.
Summary of the salient findings:
Using AFLD mice model and in vitro L02 cells, authors demonstrated the protective effect of caveolin-1 on APAP-induced hepatotoxicity. Overexpression of CAV1 alleviating hepatotoxicity and downregulated Ang II, p-EGFR/EGFR and p-ERK/ERK expression. Also, application of Ang II receptor, ERK1/2 antagonists alleviated APAP induced hepatotoxicity in AFLD.
The proposed study is very interesting, results are explained well but I have the following comments and concerns.
1. Caveolin-1 protein expression reduced in AFLD models (role in the pathogenesis of AFLD). However it’s not how caveolin 1 expression is regulated. Does it also affects caveolin-1 mRNA level ?
2. Caveolin-1 alleviates APAP induced hepatotoxicity in AFLD. There is no mention about the clinical relevance. What is the status of caveolin-1 expression in clinical settings?
3. Authors demonstrated blocking of autophagy and restoration of autophagic flux with Caveolin overexpression.
In Fig4 B, LC3II protein level & densitometry is not matching with Fig. 6E. From Fig4b WB there looks an increase in LC3II to LC3I expression in LV-ctrl + Etoh +APAP condition but densitometric analysis shows reduction which is not in line with fig6E where graph show increase in similar settings (LV-ctrl + Etoh +APAP).
4. Minor comments-
a) In line 7, there is extra space between Institute of and Innovative.
I recommend the manuscript be accepted for publication, with addressing these concerns

Round 2
Reviewer 2 Report
The authors responded to my requests.